# Cause-specific student absenteeism monitoring in K-12 schools for detection of increased influenza activity in the surrounding community—Dane County, Wisconsin, 2014–2020

**Jonathan L. Temte**[1], **Shari Barlow**[1‡], **Maureen Goss**[1‡], **Emily Temte**[1‡], **Amber Schemmel**[1‡], **Cristalyne Bell** [1‡*], **Erik Reisdorf**[2‡], **Peter Shult**[2‡], **Mary Wedig**[2‡], **Thomas Haupt**[3‡], **James H. Conway**[4‡], **Ronald Gangnon**[5], **Amra Uzicanin**[6]

1 Department of Family Medicine and Community Health, School of Medicine and Public Health, University of Wisconsin, Madison, Wisconsin, United States of America, 2 Wisconsin State Laboratory of Hygiene, Madison, Wisconsin, United States of America, 3 Wisconsin Division of Public Health, Wisconsin Department of Health Services, Madison, Wisconsin, United States of America, 4 Division of Infectious Diseases, Department of Pediatrics, School of Medicine and Public Health, University of Wisconsin, Madison, Wisconsin, United States of America, 5 Department of Biostatistics, School of Medicine and Public Health, University of Wisconsin, Madison, Wisconsin, United States of America, 6 U.S. Centers for Disease Control and Prevention, Atlanta, Georgia, United States of America

☯ These authors contributed equally to this work.
‡ SB, MG, ET, AS, CB, ER, PS, MW, TH and JHC also contributed equally to this work.
* Cristalyne.bell@fammed.wisc.edu

## Abstract

### Background

Schools are primary venues of influenza amplification with secondary spread to communities. We assessed K-12 student absenteeism monitoring as a means for early detection of influenza activity in the community.

### Materials and methods

Between September 2014 and March 2020, we conducted a prospective observational study of all-cause (a-TOT), illness-associated (a-I), and influenza-like illness–associated (a-ILI) absenteeism within the Oregon School District (OSD), Dane County, Wisconsin. Absenteeism was reported through the electronic student information system. Students were visited at home where pharyngeal specimens were collected for influenza RT-PCR testing. Surveillance of medically-attended laboratory-confirmed influenza (MAI) occurred in five primary care clinics in and adjoining the OSD. Poisson general additive log linear regression models of daily counts of absenteeism and MAI were compared using correlation analysis.

### Findings

Influenza was detected in 723 of 2,378 visited students, and in 1,327 of 4,903 MAI patients. Over six influenza seasons, a-ILI was significantly correlated with MAI in the community (r =

**Data Availability Statement:** Data can be found at Harvard Dataverse: https://doi.org/10.7910/DVN/EGHXLS.

**Funding:** This study has been supported by Centers for Disease Control and Prevention (www.cdc.gov) through cooperative agreement # 5U01CK000542-02-00. JLT received the award. CDC project officer (AU) assisted with study design, interpretation of data, writing of this report, and in the decision to submit the paper for publication. The findings and conclusions in this study are those of the authors and do not necessarily represent the official position of the Centers for Disease Control and Prevention.

**Competing interests:** JLT has received past research funding from Quidel Corporation. Quidel provided in-kind Sofia analyzers and Influenza A+B FIA tests to the Wisconsin research team. This does not alter our adherence to PLOS ONE policies on sharing data and materials. Quidel did not direct or exert any influence over study design, data collection and analysis, decision to publish, or preparation of the manuscript.

0.57; 95% CI: 0.53–0.63) with a one-day lead time and a-I was significantly correlated with MAI in the community (r = 0.49; 0.44–0.54) with a 10-day lead time, while a-TOT performed poorly (r = 0.27; 0.21–0.33), following MAI by six days.

## Discussion

Surveillance using cause-specific absenteeism was feasible and performed well over a study period marked by diverse presentations of seasonal influenza. Monitoring a-I and a-ILI can provide early warning of seasonal influenza in time for community mitigation efforts.

## Introduction

Seasonal influenza contributes to widespread individual and societal costs. Infection causes school and work absenteeism, medical visitation, hospitalization, and death, with direct and indirect costs between $6.3 and $25.3 billion annually in the United States [1]. Although school-aged children have the highest rates of medically-attended influenza (MAI) [2], they have relatively low rates of hospitalization and death [3]. Nevertheless, influenza is a primary driver of school absenteeism for students; medically-attended acute respiratory infection (ARI) due to influenza accounts for nearly 50% of school days missed due to ARI [4].

Monitoring and surveillance systems for seasonal and pandemic influenza provide situational awareness for public health interventions and health system preparedness, assessment of timing and intensity of outbreaks, and virological characterization [5]. These systems typically involve evaluations of influenza-like illness (ILI) prevalence, rates of MAI and hospitalization, results of laboratory testing, and estimates of mortality [2, 3]. Accordingly, surveillance systems are usually based in more populated areas supporting the necessary healthcare infrastructure.

Because of generally high influenza attack rates in school-aged children [2], significant linkages between MAI and school absenteeism [4], and the purported role of transmission and amplification of influenza in schools [6], use of cause-specific absenteeism may provide a useful and inexpensive complementary method for influenza monitoring and early warning of school-based as well as community-wide outbreaks. Absenteeism data are routinely collected within 14,000 US school districts [7]. A functional and generalizable system could provide wide geographic coverage for influenza monitoring efforts, particularly in resource-scarce jurisdictions.

Starting with the first report relating school absenteeism to an influenza outbreak in 1958 [8], there have been numerous attempts to evaluate school absenteeism for detecting accelerated spread of seasonal and pandemic influenza within communities [9–14]. For example, comparisons performed during the 2009 influenza A(H1N1) pandemic provided promisingly high correlation (r = 0.92) between ILI-related absenteeism and weekly counts of hospitalized influenza cases [14]. The autumnal pandemic outbreak, however, provided very strong signals in both absenteeism and hospitalization which are not representative of typical outbreaks of seasonal influenza.

The goal of the ORegon CHild Absenteeism due to Respiratory Disease Study (ORCHARDS) is to develop and implement a replicable system for daily school-based monitoring of ILI-specific student absenteeism in kindergarten through grade 12 (K-12) schools, and to assess the system's usability for early detection of accelerated influenza and other

respiratory pathogen transmission in schools and surrounding communities [15]. In this paper, we discuss the relationship between student absenteeism and MAI in surrounding communities.

## Materials and methods

ORCHARDS is a prospective, community-based study that captures school absenteeism data on a daily basis. In addition, school-aged children with ARI are recruited to participant in a home study which allows evaluation of demographic, clinical, and virological correlates of respiratory infections. The complete, detailed methodology for ORCHARDS has been published elsewhere [15]. Briefly, ORCHARDS is based in the Oregon School District (OSD: www.oregonsd.org), which includes the villages of Oregon and Brooklyn, located in a semi-rural area of Dane County, Wisconsin. The district comprises three elementary schools (grade K-4)—two in Oregon and one in Brooklyn—with 1,503 students, one intermediate school (grade 5–6) with 623 students, one middle school (grade 7–8) with 596 students, and one high school (grade 9–12) with 1,145 students, all located in Oregon [16].

Data collection for this study commenced with the start of the 2014–2015 academic year, on September 2, 2014, and concluded with the closure of OSD schools related to the emerging SARS-CoV-2 pandemic after March 13, 2020. Data collection occurred over six academic years and six sequential influenza seasons, marked by significant diversity in the dominant influenza types and subtypes, outbreak intensity, and timing of outbreaks relative to the academic calendar (Table 1).

### Absenteeism reporting system

OSD utilizes Infinite Campus® (https://www.infinitecampus.com), a commercially available electronic student information system, to track attendance. Parents/guardians report their child's absences to school attendance staff by calling an automated telephone system. Callers are prompted to provide the student's name and reason for absence, including symptoms if the child has a cold or flu-like illness. Uniform messaging is provided on each school's absentee lines with additional information pertaining to ILI symptoms:

> *"Please inform us if your child has any flu-like symptoms such as fever with cough, sneezing, chills, sore throat, body aches, fatigue, runny nose, and/or stuffy nose."*

Subsequently, school attendance staff members identify a student and record a reason for absence from a modifiable, drop-down pick list. To facilitate this study, the OSD Information

**Table 1. Dominant influenza types and subtypes across six influenza seasons, along with outbreak severity and timing.**

|  | 2014–2015 | 2015–2016 | 2016–2017 | 2017–2018 | 2018–2019 | 2019–2020 |
|---|---|---|---|---|---|---|
| **Dominant influenza type/subtype**[1] | A(H3) | A(H1) | A(H3) | A(H3) | A(H1) A(H3) | B (early) A(H1) (late) |
| **Other influenza in circulation**[1] | late B | with B | with B | with B | minimal B | minimal A(H3) |
| **Outbreak severity**[1] | moderately severe | low | moderate | severe | moderate | moderately severe |
| **Wisconsin Outbreak peak timing**[2] | Dec/Jan | Mar | Feb | Jan/Feb | Mar | Feb |

[1]Characterization of influenza season based on data presented by Centers for Disease Control and Prevention. FluView—Weekly U.S. Influenza Surveillance Report. Accessed 10/16/2020 at: https://www.cdc.gov/flu/weekly/index.htm.

[2]Wisconsin outbreak timing based on peak PCR detections reported by Wisconsin State Laboratory of Hygiene. Influenza Activity. Accessed 11/23/2020 at: http://www.slh.wisc.edu/wcln-surveillance/surveillance/virology-surveillance/influenza-activity/.

Technology (IT) staff added "Absent due to influenza-like illness" (a-ILI) to existing categories of student absenteeism in Infinite Campus®.

For this study, we define absenteeism for an individual student as being absent for the entire day or any part of the day. This definition provides simplicity and generalizability for electronic data retrieval. All-cause or total absenteeism (a-TOT) is defined as an absence for any reason. Absence due to illness (a-I) is an absence due to any reported illness. Absence due to ILI (a-ILI) is a subset of a-I for which ILI symptoms are reported by the parent/caregiver when calling the school absenteeism telephone line. We considered several established definitions for ILI [17, 18]. For ease of recognition by school attendance staff without medical backgrounds, however, ILI was defined as the presence of fever and at least one respiratory tract symptom (cough, chest congestion, sore throat, scratchy throat, sneezing, runny nose, nasal discharge, nasal stuffiness, or nasal congestion).

OSD IT staff created automated processes within Infinite Campus® to extract daily counts of absent students by school, grade, and absence type (a-TOT, a-I, and a-ILI). The data contain no personal identifiers and are fully compliant with the Family Educational Rights and Privacy Act (FERPA: 20 U.S.C. § 1232g; 34 CFR Part 99) [19]. The data are sent to a secure file transfer protocol (ftp) site at the University of Wisconsin Department of Family Medicine and Community Health, as illustrated in Fig 1.

## Laboratory assessment of influenza and other respiratory viruses

The full details pertaining to recruitment of students for data and specimen collection at home, including inclusion and exclusion criteria, are available in the ORCHARDS methods paper [15]. Inclusion criteria included: (1) student attends, or is eligible to attend (e.g., home schooled), a school within the OSD; (2) has an illness characterized by ≥2 of 6 acute respiratory infection/ILI symptoms (nasal discharge; nasal congestion; sneezing; sore throat; cough; fever); and (3) scores ≥2 points on the Jackson scale. Exclusion criteria included: (1) illness onset ≥7 days before anticipated time of specimen collection; (2) anatomical defect for which nasal specimen collection is contraindicated; and (3) student participated too recently (<7 days during peak influenza period and <30 days during other times, as determined by medically-attended surveillance program).

Laboratory-supported data to evaluate the role of influenza on a-ILI status were collected for the OSD students who opted to participate in ORCHARDS [15]. Briefly, 2,378 home visits were conducted during the study period for children who met criteria for ARI. Absenteeism was not required for a home visit to occur. A nasal swab specimen was obtained for testing using the Quidel Sofia Influenza A+B fluorescent immunoassay (https://www.quidel.com/immunoassays/rapid-influenza-tests/sofia-influenza-fia) by ORCHARDS staff. In addition, a high oropharyngeal or nasopharyngeal swab was collected for testing with influenza RT-PCR [in-vitro diagnostic (IVD) FDA-approved CDC Human Influenza Virus Real-time RT-PCR Diagnostic Panel (Cat.# FluiVD03)] [20] and a respiratory pathogen panel [21] at the Wisconsin State Laboratory of Hygiene (WSLH).

## Assessment of community MAI

An independent and long-standing influenza surveillance system was used to assess MAI in and around the OSD. The Optional Influenza Surveillance Enhancements/Wisconsin Influenza Incidence Surveillance Project is sponsored by the Centers for Disease Control and Prevention (CDC) and conducts prospective, active surveillance for influenza and other respiratory viruses in medically-attended patients with ARI or ILI. The ORCHARDS research team organizes this surveillance system [2, 22]. Active surveillance occurs at five family medicine clinics within or

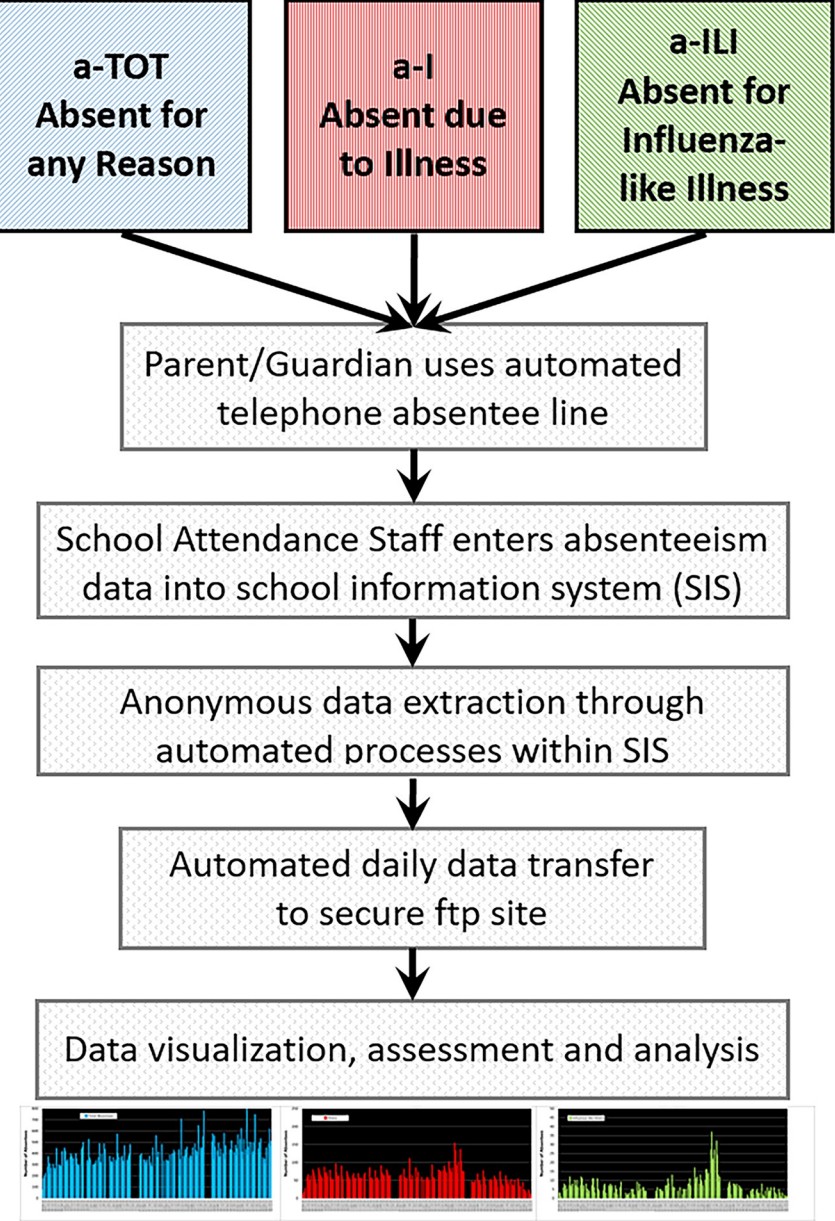

**Fig 1. Flow diagram of absenteeism data from telephone reporting by parents/guardians, to entry into the student information system at the Oregon School District, to data transfer to the ORCHARDS research team.**

adjoining the OSD. Specific clinic locations include Belleville, Oregon, Madison (2 sites), and Verona (Fig 2). Together, the clinics recorded 412,752 ambulatory visits between June 29, 2014, and June 29, 2019, and assessed 4,903 individual ARI patients. Each assessed ARI patient was tested using an on-site rapid influenza diagnostic test, and by influenza RT-PCR [20] and a multiplexed PCR respiratory pathogen panel [21] performed at WSLH.

## Data analyses

We assessed the relationship between a-ILI status (outcome) and influenza (exposure) in the subset of students who had a home visit and who could be absent from school, thus excluding

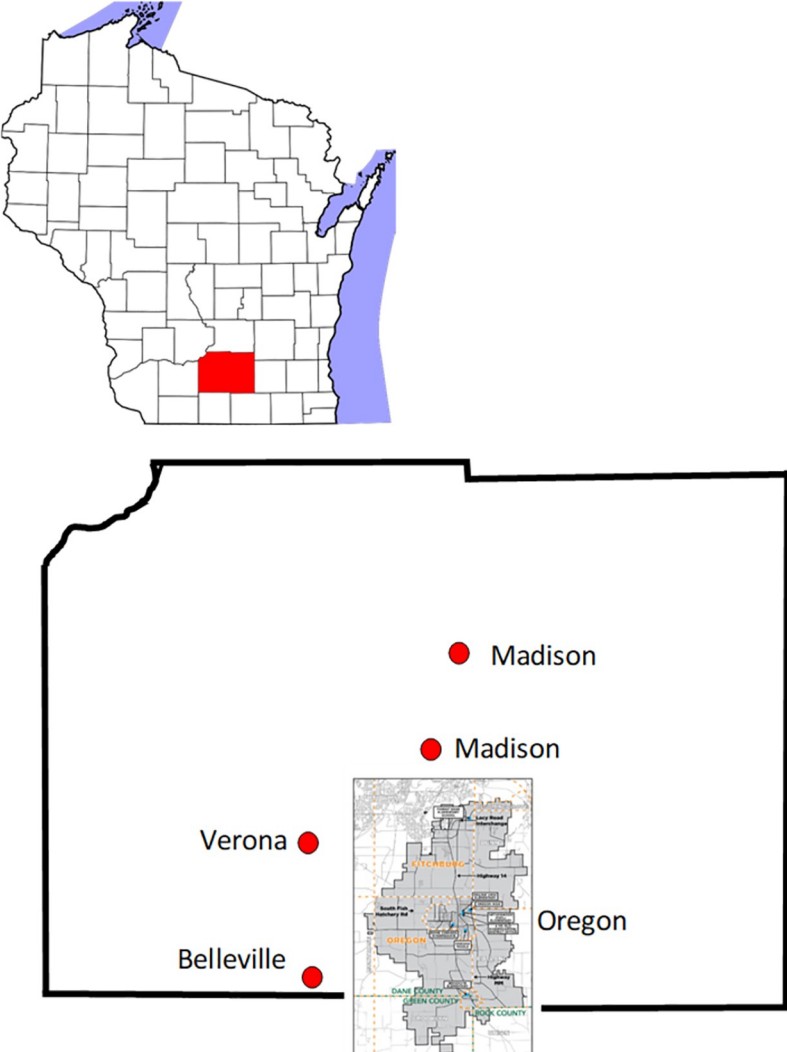

**Fig 2. Location of the Oregon School District (OSD) in the villages of Oregon and Brooklyn in South Central Wisconsin.** The red dots depict locations of family medicine clinics that participate in long-term surveillance of medically-attended influenza.

students who were home-schooled or had illness episodes on weekend days, vacation days, school closure days, or during the summer. Students were classified as a-ILI if they were absent and had fever plus one respiratory tract symptom (cough, sore throat, nasal congestion, or runny nose) as reported during the home visit. The odds ratio for meeting the a-ILI definition given laboratory-confirmed influenza, compared to a-ILI occurring in the absence of influenza was calculated using standard logistic regression methods.

A generalized additive quasi-Poisson log-linear regression model was fit to the daily counts of MAI from surrounding surveillance clinics as a function of calendar date represented using a thin-plate regression spline and day of the week represented using dummy variables [23, 24]. Similarly, generalized additive quasi-Poisson log-linear regression models were fit to daily counts of three absenteeism categories (a-TOT, a-I and a-ILI) as a function of calendar date

represented using a thin-plate regression spline and day of the week represented using dummy variables.

Sample cross-correlation functions were calculated between the (square-root) daily number of medically attended influenza cases and lagged (or leading) (square-root) daily absenteeism counts from three sources (a-TOT, a-I, a-ILI) for each school year individually and for all school years combined. The optimal lag (or lead) resulting in the maximum cross-correlation was identified for each absenteeism series.

### IRB and project oversight

All components of this proposed study were reviewed and approved by the Human Subjects Committees of the Education and Social/Behavioral Sciences IRB (initial approval on September 4, 2013; ID number: 2013–1268) and the University of Wisconsin Health Sciences-IRB (initial approval on December 5, 2013, with additional approvals as the protocol expanded and modified; ID number: 2013–1357). The study is in full compliance with the Health Insurance Portability and Accountability Act of 1996 (HIPAA), FERPA, and all other federally mandated human subjects regulations. The US Office of Management and Budget has approved all forms used in this study.

Written informed consent was obtained from parents/guardians of minor students and from students aged ≥18 years during home visits for collection or data and respiratory specimens as detailed in the ORCHARDS methodology [15]. All absenteeism data from the OSD's information system was anonymous; no informed consent was needed per review by the Human Subjects Committees of the Education and Social/Behavioral Sciences IRB.

Per the cooperative agreement, the CDC project officer (AU) assisted with study design, interpretation of data, writing of this report, and in the decision to submit the paper for publication.

## Results

### Absenteeism

We monitored 3,738,562 potential student-days over six academic years. During this period, enrollment at the OSD slowly increased from 3,588 to 3,887 students (Table 2). Absenteeism was relatively stable over six years with a-TOT, a-I and a-ILI accounting for 357,054 (9.6%), 68,809 (1.8%) and 8,161 (0.22%) of total potential student days, respectively. Percentages for a-I and a-ILI in 2019–2020 were elevated due to the abrupt and early end of in-person instruction and absence recording after March 13, 2020 during the influenza season. Categories of recorded absenteeism had varying relationships with grade, with a-TOT increasing (r = 0.924; P<0.001) and a-ILI declining with grade level (r = -0.945; P<0.001). No such effects were seen for a-I (r = -0.498; P = 0.083) (Fig 3).

### Medically attended influenza

Throughout the study period, 4,903 ARI patients were evaluated for influenza at the five surveillance clinics (Table 3). MAI was confirmed by PCR for 1,327 (27.1%) individual patient encounters, with influenza A noted in 943 (19.2%) and influenza B present in 385 (7.9%) encounters. One adult patient was positive for both influenza A and B. The prevalence of influenza in ARI patients and the relative distribution of influenza types varied across the six-year study period. The percentage of ARI patients with MAI varied between 17.9% in 2015–2016 and 39.6% in 2019–2020. Influenza B comprised between 5.4% of MAI in 2018–2019 and 48.4% in 2017–2018.

**Table 2. School enrollment estimates and student absenteeism by defined type\* and school year—Oregon School District, Dane County, Wisconsin: 2014–2020.**

|  | 2014–2015 | 2015–2016 | 2016–2017 | 2017–2018 | 2018–2019 | 2019–2020[†] |
|---|---|---|---|---|---|---|
| **SCHOOL ENROLMENT** | | | | | | |
| **Total Estimated Enrollment\*\*** | 3,588 | 3,713 | 3,749 | 3,828 | 3,867 | 3,887 |
| **Total Curricular Days** | 176 | 177 | 175 | 176 | 174 | 123 |
| **Total Possible Student Days** | 629,024 | 635,076 | 649,775 | 673,728 | 672,858 | 478,101 |
| **STUDENT ABSENTEEISM** | | | | | | |
| **a-TOT Days (% of total)** | 56,477 (8.98%) | 55,871 (8.80%) | 56,511 (8.70%) | 63,476 (9.42%) | 69,092 (10.26%) | 55,627 (11.63%) |
| *Average Daily a-TOT (std. dev.)* | *320.9 (78.4)* | *315.7 (79.2)* | *322.9 (80.4)* | *360.7 (96.3)* | *397.1 (106.2)* | *452.3 (118.0)* |
| **a-I Days (% of total)** | 12,024 (1.91%) | 12,158 (1.91%) | 11,922 (1.83%) | 11,710 (1.74%) | 10,312 (1.53%) | 10,683 (2.23%) |
| *Average Daily a-I (std. dev.)* | *68.3 (39.0)* | *68.7 (27.0)* | *68.1 (27.0)* | *66.5 (30.1)* | *59.3 (21.7)* | *86.9 (39.0)* |
| **a-ILI Days (% of total)** | 1,314 (0.21%) | 1,609 (0.25%) | 1,309 (0.20%) | 1,261 (0.19%) | 1,141 (0.17%) | 1,527 (0.32%) |
| *Average Daily a-ILI (std. dev.)* | *7.5 (7.3)* | *9.1 (6.6)* | *7.5 (5.9)* | *7.2 (5.8)* | *6.6 (5.5)* | *12.4 (13.7)* |

\*Defined types of absenteeism: a-TOT = absent for any reason; a-I = absent due to illness; a-ILI = absent due to influenza-like illness.

\*\*Total estimated enrollment excluding pre-kindergarten students (for whom absenteeism data is not submitted). Data extracted from: Wisconsin Department of Public Instruction, Wisconsin Information System for Education data Dashboard: https://wisedash.dpi.wi.gov/Dashboard/portalHome.jsp.

[†]Early termination of the school year due to SARS-CoV-2 on March 13, 2020.

There were 386 (29.1%) MAI cases in school-aged children (ages 5–18 years) and 941 (70.9%) MAI cases in younger children (ages 0–4 years) and adults (ages ≥19 years) combined. The mean dates of MAI cases were similar between the school-aged children and other individuals. A significant difference in timing (Mann-Whiney; P = 0.031) was noted for only one season (2014–2015), during which MAI in school-aged children occurred 16 days later than for other individuals.

## Timing of absenteeism and MAI

The relative timing of a-TOT, a-I, a-ILI and MAI over the entire six-year time series are provided in Fig 4. Longer breaks (e.g., winter, spring, and summer) are easily seen in the graphic. Total absenteeism (a-TOT) shows less day-to-day variability than a-I and a-ILI. As expected, MAI demonstrates distinct seasonality, with influenza cases primarily occurring in the winter and early spring.

## Relationship between influenza detection and a-ILI status

The demographics of the students participating in home visits is presented in Table 4.

Of 2,378 home visits conducted with ORCHARDS participants, 2,143 (90%) occurred at times when absenteeism was possible. Of those, 1,214 (57%) met the criteria for a-ILI and 639 (30%) students had laboratory-confirmed influenza (Table 5). There was a significant association between a-ILI and influenza ($X^2$ = 244.2; d.f. = 1; P<0.0001) and the odds ratio for meeting a-ILI criteria if influenza was detected was 5.52 (95% CI: 4.40–6.93).

## Relationship between absenteeism and MAI

For all academic years/influenza seasons combined, a-TOT was positively correlated with MAI with a maximum correlation of 0.27 (95% CI: 0.21–0.33) achieved when a-TOT followed MAI by six days (Fig 5); the maximum correlation with a-TOT preceding MAI was 0.21 (0.15–0.27) with a lead of one day. a-I had a stronger positive correlation with MAI, with a maximum correlation of 0.49 (0.44–0.54) when leading MAI by 10 days. a-ILI had the strongest positive

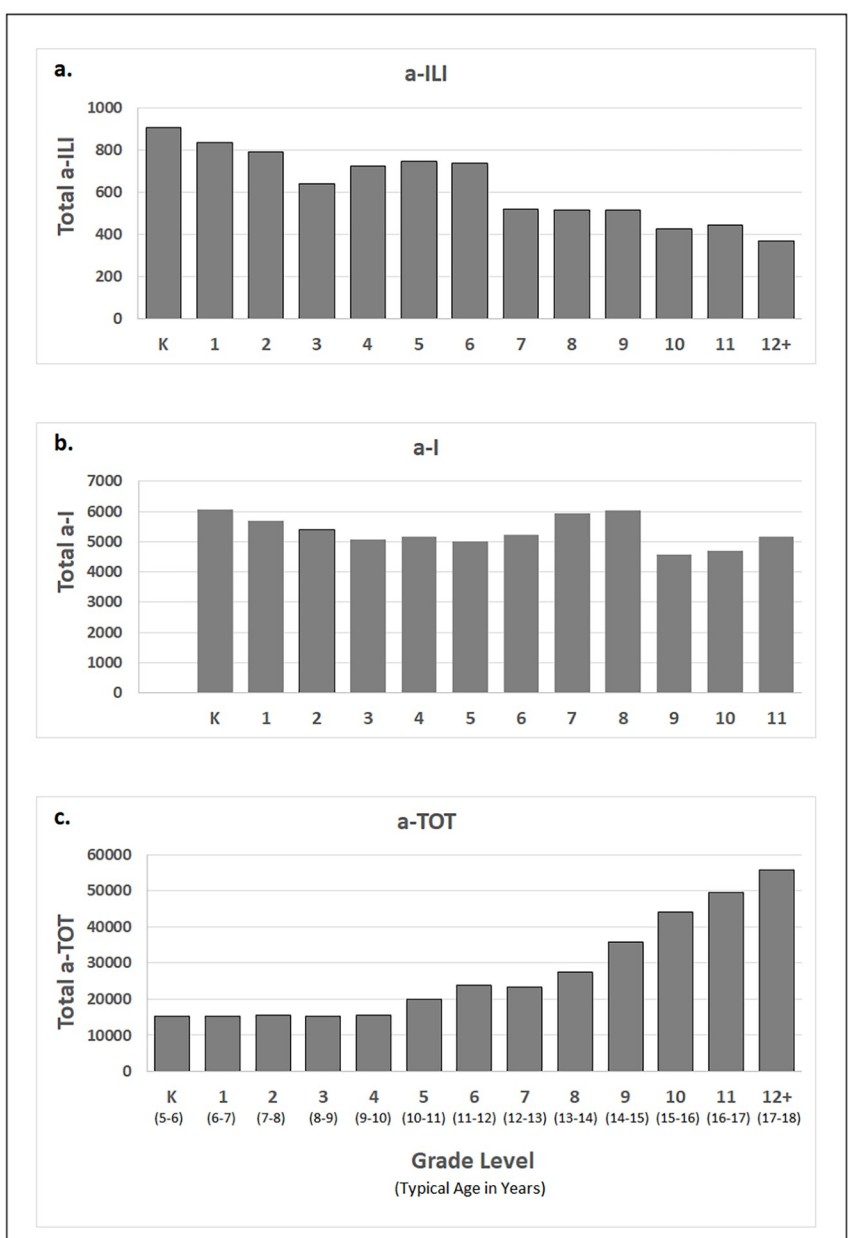

**Fig 3. Total absentee counts, by grade level, across entire study for a-ILI, a-I and a-TOT.** Defined types of absenteeism: a-TOT = absent for any reason; a-I = absent due to illness; a-ILI = absent due to influenza-like illness.

correlation with MAI, with a maximum correlation of 0.57 (0.53–0.63) when leading MAI by one day.

For individual school years/influenza seasons, the maximum correlation (lagging or leading) between a-TOT and MAI varied from 0.20 (0.05–0.34) when trailing MAI by 11 days in 2017–2018 to 0.39 (0.23–0.47) when trailing MAI by six days in 2019–2020. When restricted to evaluations for a-TOT preceding MAI only, the range was 0.09 (-0.06–0.23) with a lead of four days in 2017–2018 to 0.39 (0.23–0.53) with a lead of eight days in 2019–2020. The maximum correlation between a-I and MAI varied from 0.34 (0.21–0.47) while preceding MAI by 13

**Table 3. Acute respiratory infection (ARI) cases evaluated and numbers of influenza A and B cases detected across six influenza surveillance clinics,[*] Dane County, Wisconsin.**

| | 2014–2015 | 2015–2016 | 2016–2017 | 2017–2018 | 2018–2019 | 2019–2020[†] |
|---|---|---|---|---|---|---|
| **All Ages** | | | | | | |
| **ARI Cases Evaluated** | 986 | 698 | 708 | 1,033 | 645 | 833 |
| **Influenza A[**]** | 170 | 106 | 95 | 230 | 141 | 201 |
| **Influenza B[**]** | 78 | 19 | 46 | 102 | 8 | 132 |
| **Total Influenza (% of ARI)** | 248 (25.2%) | 125 (17.9%) | 141 (19.9%) | 332 (32.1%) | 149 (23.1%) | 332 (39.9%) |
| **Mean date of influenza** | 1/25/2015 | 3/04/2016 | 3/02/2017 | 2/06/2018 | 2/28/2019 | 2/07/2020 |
| **Age 5–18 years** | | | | | | |
| **ARI Cases Evaluated (% of all ages)** | 157 (15.9%) | 132 (18.9%) | 142 (20.1%) | 225 (21.8%) | 130 (20.2%) | 236 (28.3%) |
| **Influenza A[**]** | 37 | 17 | 29 | 59 | 42 | 50 |
| **Influenza B[**]** | 12 | 7 | 14 | 39 | 4 | 76 |
| **Total Influenza (% of ARI)** | 49 (31.2%) | 24 (18.2%) | 43 (30.3%) | 98 (43.6%) | 46 (35.4%) | 126 (53.4%) |
| **Mean date of influenza** | 1/28/2015 | 3/03/2016 | 3/03/2017 | 2/07/2018 | 2/28/2019 | 2/07/2020 |
| **Ages 0–4 years and ≥19 years** | | | | | | |
| **ARI Cases Evaluated (% of all ages)** | 829 (84.1%) | 566 (81.1%) | 566 (79.9%) | 808 (78.2%) | 515 (79.8%) | 597 (71.7%) |
| **Influenza A[**]** | 133 | 89 | 66 | 171 | 99 | 151 |
| **Influenza B[**]** | 66 | 12 | 32 | 63 | 4 | 56 |
| **Total Influenza (% of ARI)** | 199 (24.0%) | 101 (17.8%) | 98 (17.3%) | 234 (29.0%) | 103 (20.0%) | 206 (34.5%) |
| **Mean date of influenza** | 1/12/2014 | 3/08/2015 | 3/01/2016 | 2/04/2017 | 2/27/2019 | 2/06/2020 |

Data are shown for all ages, school-aged children (age 5–18 years) and non-school-aged individuals (ages 0–4 and ≥19 years). Mean dates of medically attended influenza are provided for each group and in season.

[*]All surveillance clinics are family medicine clinics affiliated with the University of Wisconsin and provide care to patients of all ages.

[**]Influenza type is based on testing using CDC Human Influenza Virus Real-time RT-PCR Diagnostic Panel [20].

[†]Clinic visits through March 20, 2020.

days in 2015–2016 to 0.71 (0.61–0.79) while preceding MAI by seven days in 2019–2020. When restricted to evaluations for a-I preceding MAI only, the range was the same. Finally, maximum correlation between a-ILI and MAI varied between 0.48 (0.36–0.58) while trailing MAI by three days in 2015–2016 and 0.82 (0.75–0.87) while trailing MAI by seven days in 2019–2020. When restricted to evaluations for a-ILI preceding MAI only, the range was 0.46 (0.33–0.57) with a lead of one day in 2015–2016 to 0.80 (0.73–0.86) with a lead of one day in 2019–2020. For all six seasons, however, significant, positive cross-correlations were achieved for a-I and a-ILI at least 14 days in advance of MAI (Fig 5). Moreover, in every season, cross-correlations with MAI were higher for a-I and a-ILI than for a-TOT; in all but 2014–2015, cross-correlations with MAI were higher for a-ILI than for a-I.

Dividing a-ILI data into two subsets, namely grades K-4 and grades 5–12, did not improve correlations. For all school years/influenza seasons combined, maximal correlations for a-ILI (K-4) and a-ILI (5–12) were 0.52 and 0.58 at lags, respectively.

## Discussion

Cause-specific daily K-12 absenteeism within a school district mirrors MAI in surrounding communities. Absenteeism due to ILI (a-ILI) and absenteeism due illness (a-I) demonstrated strong and moderate positive correlations with MAI, respectively, over a period of six consecutive influenza seasons. The maximal correlations either led (a-I) or occurred coincidentally (a-ILI) with MAI. Moreover, as time frames are shifted, significant correlations are demonstrated between cause-specific absenteeism (a-ILI and a-I) and MAI well in advance of MAI in all

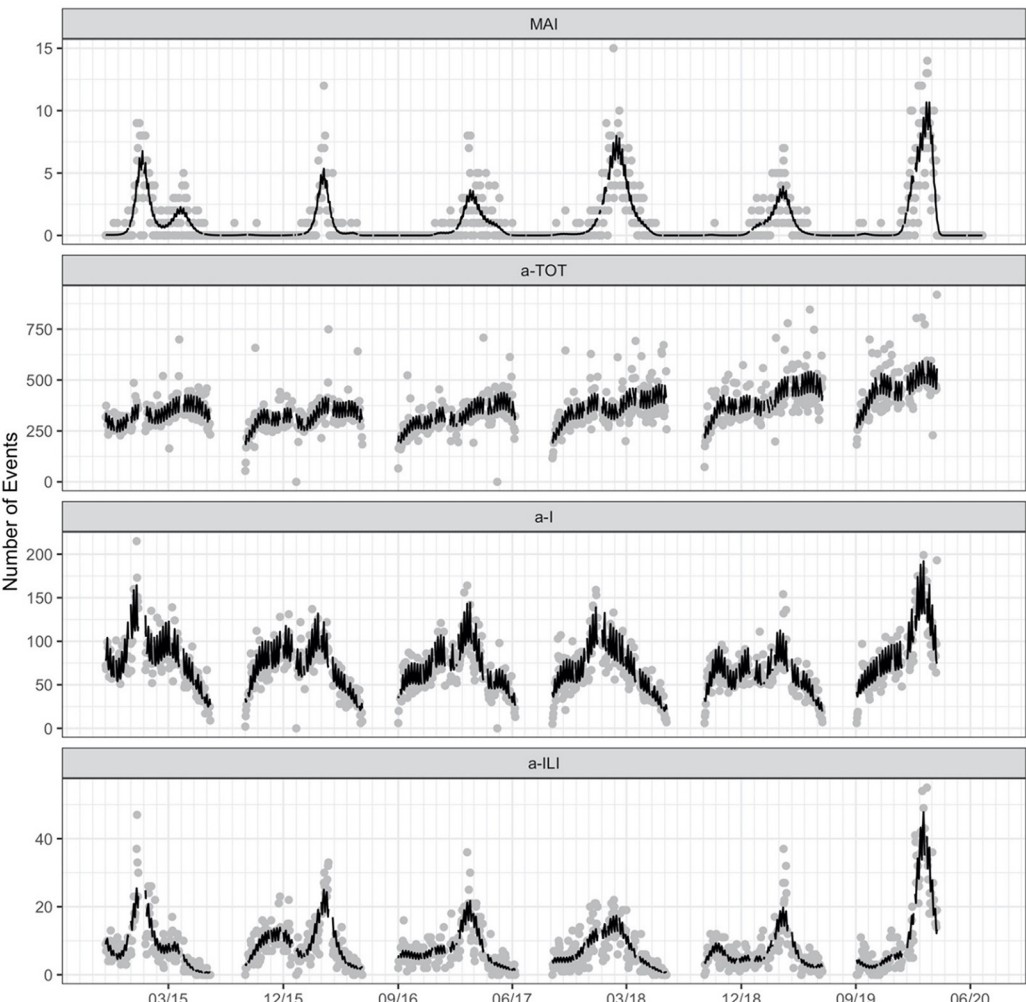

**Fig 4. Numbers of daily student absences in each of three categories (a-TOT = absent for any reason, a-I = absent due to illness and a-ILI = absent due to influenza-like illness) and number of all-age medically-attended influenza (MAI) cases in surrounding communities during six school years/influenza seasons.** The black line is the estimated mean number of daily events from a generalized additive quasi-Poisson regression model with calendar date (thin-plate spline) and day of week (dummy variables) as predictors. Gaps in the fitted curves represent breaks in the school calendar.

**Table 4. Characteristics of 2,378 ORCHARDS student participants in the home study.**

| | |
|---|---|
| Age, mean ± sd; [range] | 9.9 ± 3.5 years [4–18] |
| Grades K-4 (4–9 years) | 1,202 (50.6%) |
| Grades 5–6 (10–11 years) | 418 (17.6%) |
| Grades 7–8 (12–13 years) | 352 (14.8%) |
| High School (14–18 years) | 406 (17.1%) |
| Sex | |
| Female | 1,024 (43.2%) |
| Male | 1,349 (56.8%) |
| Time from symptom onset to home visit | 53.8 ± 32.3 hours |

**Table 5. Relationship between influenza laboratory detection and absence with influenza-like illness (a-ILI) in a cohort of students visited at home.**

| Exposure | | Outcome | | |
|---|---|---|---|---|
| | | a-ILI | Not a-ILI | Total |
| | Influenza (+) | 526 | 113 | 639 |
| | Influenza (-) | 688 | 816 | 1,504 |
| | Total | 1,214 | 929 | 2,143 |

Oregon School District, Dane County, Wisconsin. Chi-square for association = 244.2 (d.f. = 1; P<0.001). Odds Ratio for meeting a-ILI criteria with exposure to influenza = 5.521 (95% CI: 4.399–6.929).

cases. Accordingly, results from this six-year study support the hypothesis that cause-specific school absenteeism monitoring can provide timely and potentially advance identification of community outbreaks of seasonal influenza. As more stringent criteria are applied to evaluate absenteeism (e.g., a-I and a-ILI), there are increasing correlations with community levels of influenza. Accordingly, a-ILI may be seen as having a relatively higher specificity for community outbreaks of influenza. A less specific criterion (a-I) may be more sensitive for influenza with a composite time lag for the maximum correlation estimated at minus six days. In general, all-cause absenteeism (a-TOT) performed less well for identifying influenza in the community.

The absenteeism monitoring tool used in ORCHARDS [15] has performed well since inception, retrieving and transmitting data from 1,001 consecutive school days with minimal maintenance. Given that absenteeism is widely monitored across school districts, student information systems are commonly employed in the K-12 environment, and a simple enhancement allowed for accurate ILI data capture and anonymized electronic file transfer, this approach may be generalizable to many of the nearly 14,000 school districts in the United States [8]. Widespread use of K-12 school-based absenteeism monitoring may augment MAI surveillance where available and could fill in geospatial gaps where MAI surveillance is not available. The composite of all grades (K-12) appeared to perform better than subsets of younger or older children.

The strength of correlations for individual outbreaks of influenza appears to follow the intensity of the influenza season. The most significant influenza seasons experienced by the OSD and the surveillance clinics (2017–2018 and 2019–2020) resulted in very high maximal correlations (r = 0.64 and 0.82, respectively) between a-ILI and MAI. This supports potential benefit of cause-based absenteeism in monitoring systems for pandemic influenza.

There have been multiple efforts to assess the utility of monitoring absenteeism of children as an indicator of influenza in the community [9–14, 25, 26]. Many reports have been based on all-cause absenteeism [9–11, 25, 26]. Whereas monitoring all-cause absenteeism is appealing and intuitive due to the routine collection of this metric, significant problems abound due to "noise" within a-TOT resulting from the multi-factorial nature of absenteeism [9, 26]. On any given day, an estimated 7.8% of American school children, or 4.3 million children, are absent [27]. Some are absent due to illness; a subset of these ill children are absent due to respiratory infections. As a-ILI comprised between 1.7–2.9% of total annual absenteeism in our study, the signal from influenza may be obscured. The Pennsylvania Influenza Sentinel School Monitoring System illustrated this concept during the 2009 influenza pandemic [9]. Poor correlation (r = 0.10; P = 0.56) was noted between all-cause absenteeism and statewide laboratory-confirmed influenza. For ORCHARDS, a-TOT performed poorer than a-I or a-ILI, with an overall maximal correlation of 0.27, and lagging behind community MAI by six days.

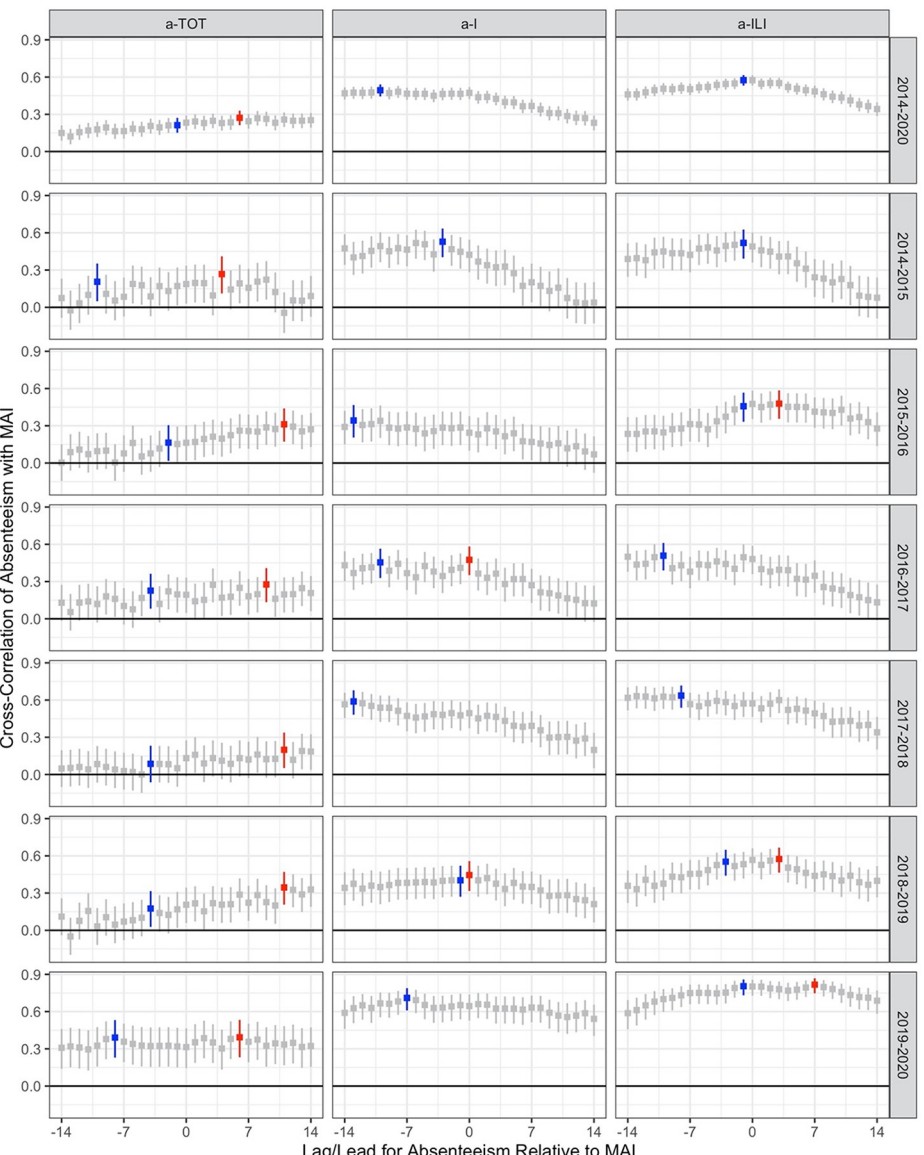

**Fig 5. Cross-correlation functions (estimate and 95% confidence interval) for square-root daily absenteeism counts and square-root medically-attended influenza (MAI) counts for all six school years/influenza seasons combined (top row) and for each school year/influenza season.** Results are shown for total absenteeism (a-TOT; first column), absenteeism due to illness (a-I; second column) and absenteeism due to influenza-like illness (a-ILI; third column). Negative lag/lead indicates absenteeism preceding MAI, while positive lag/lead indicates MAI preceding absenteeism. The maximal correlation for absenteeism preceding MAI is highlighted in blue; if different, the maximum correlation over the entire time frame (-14 days to +14 days) is highlighted in red.

There are very few examples of utilization of cause-specific absenteeism for detection of influenza. One study from the 2009 influenza pandemic demonstrated poor (r = 0.33), high (r = 0.63), and very high (r = 0.92) correlation between confirmed influenza hospitalizations and all-cause absenteeism, ILI health office visits, and ILI-related absenteeism, respectively [14]. Likewise, ORCHARDS demonstrated lower correlation for a-TOT (r = 0.39) and very high correlation for a-ILI (r = 0.82) during 2019–2020, a year characterized by a prolonged and severe seasonal influenza outbreak. The combination of multiple years in ORCHARDS

provides a more robust assessment of cause-specific absenteeism. In all years, a-I demonstrated higher maximal correlation than a-TOT; in all but one school year a-ILI demonstrated higher correlation than a-I.

We are aware of a single example of the use of automated reporting of absenteeism data for influenza monitoring. A smart card reader was employed to capture attendance at 107 schools in Hong Kong, allowing for near real-time reporting [28]. Implementation of this system, using routinely-collected information, was found to be feasible and acceptable. This study also reported a modest maximal cross-correlation coefficient between ILI-specific absenteeism and hospital laboratory-confirmed influenza (CCC = 0.434) with a three-week lag. The results, however, are not directly comparable to ORCHARDS due to differences in ascertainment of a-ILI status and use of hospital vs. outpatient assessment of MAI.

Following cessation of in-person instruction due to the SARS-CoV-2 pandemic, the OSD did not monitor student absenteeism between March 16, 2020 and the end of the academic year on June 5, 2020. Absenteeism monitoring resumed on September 9, 2021, with a hybrid teaching model. An initial analysis of data available suggests that COVID-19-related absenteeism may provide an early warning of SARS-CoV-2 activity in the surrounding community.

Our study had several limitations. First, ORCHARDS is limited to a single, relatively small school district that is more affluent, less diverse, and better educated than the US average [29]. In addition, OSD parents were engaged with this study and many of the study personnel have long-term involvement within the community. Second, OSD had an existing telephone absenteeism reporting system with messaging dating back to the 2009 influenza pandemic. Accordingly, parents and guardians were attuned to reporting respiratory symptoms. The transition to ILI reporting was a small change in usual procedure. Third, OSD had sufficient IT staff skills to allow programming for autonomous data reporting. Fourth, we used a modified ILI definition for ease of recognition by the attendance staff. Although different from other ILI definitions [17, 18], a-ILI as defined in ORCHARDS was highly associated with laboratory-confirmed influenza [15]. Fifth, absenteeism monitoring is only an option during times when schools are in session. As a consequence, gaps may occur during planned (e.g., winter and spring breaks) and unplanned breaks (e.g., weather closure) in the academic calendar. Winter and spring breaks are no longer than 16 and 9 days, respectively; unplanned closures tend to be of short (1–3 day) duration. Unusually timed influenza outbreaks during the long (3 month) summer break would be missed by absenteeism monitoring.

This study also has notable strengths. First, there have been excellent community acceptance and participation in all aspects of ORCHARDS as evidenced by students participating in 2,378 home visits, thus allowing for validation of the a-ILI as a marker of influenza. Second, reporting of absenteeism into the telephone system by parents/guardians has not decreased over time. Third, ORCHARDS is a longitudinal, multiyear study. This extended, six-year time frame encompasses a diversity of influenza in terms of type and subtype, proportionality of types, seasonal intensity, and timing of influenza activity relative to the academic calendar. Moreover, results in ORCHARDS were not biased by the significant signal emanating from an influenza pandemic. Fourth, ORCHARDS was enhanced by the existence of a longstanding and highly functional clinic-based MAI surveillance system based within and surrounding the OSD. Finally, we utilized a robust statistical method to assess the patterns of MAI and absenteeism, comparing daily levels across six influenza seasons.

In conclusion, cause-specific absenteeism monitoring in K-12 schools identified community influenza outbreaks with high reliability across six distinct and diverse influenza seasons. The timing of periods of increased absenteeism either preceded or coincided with MAI in surrounding communities. The fully automated monitoring system was easily implemented and maintained and performed consistently over a six-year period. Accordingly, school-based,

cause-specific and near real-time absenteeism monitoring can contribute to the timely detection of influenza in the community, thus allowing for mitigation efforts, such as use of face masks, physical distancing, testing, isolation, quarantining, and enhanced vaccination programs. As school absenteeism monitoring is a common feature of schools and school districts across the United States, this approach has the potential to enhance influenza surveillance efforts, especially in low-resource areas, and provide early warnings for outbreak identification and containment.

## Acknowledgments

We would like to acknowledge Dr. Yenlik Zheteyeva, MD, MPH, formerly of CDC's Division of Global Migration and Quarantine, for her contributions to the study design and implementation. We are also grateful to Ms. Ashley Fowlkes, MPH, of CDC's Influenza Division for her assistance in accessing and interpreting the OISE/IISP data for the purpose of this study. We thank Cecilia He, Caroline Hamer, Bradley Maerz, Lily Comp, Mitchell Arnold, Kimberly Breunig, and Sarah Clifford for their assistance with data collection and Rich Griesser, Tim Davis, Tonya Danz, and Erika Hanson (Virology Laboratory Staff at the Wisconsin State Laboratory of Hygiene) for their assistance with specimen testing throughout ORCHARDS. Special thanks for Dr. Brian Bussler, Superintendent, Jon Tanner, Information Technology Director, and the attendance staff at OSD. Finally, a multitude of student participants of ORCHARDS and their families made this research possible.

## Author Contributions

**Conceptualization:** Jonathan L. Temte, Shari Barlow, Amra Uzicanin.

**Data curation:** Maureen Goss, Emily Temte, Amber Schemmel, Cristalyne Bell.

**Formal analysis:** Ronald Gangnon.

**Funding acquisition:** Jonathan L. Temte.

**Methodology:** Jonathan L. Temte, Shari Barlow, Maureen Goss, Emily Temte, Amber Schemmel, James H. Conway, Amra Uzicanin.

**Project administration:** Shari Barlow, Maureen Goss, Emily Temte, Amber Schemmel, Cristalyne Bell.

**Resources:** Erik Reisdorf, Peter Shult, Mary Wedig, Thomas Haupt.

**Software:** Amber Schemmel.

**Supervision:** Jonathan L. Temte, Shari Barlow.

**Validation:** Maureen Goss, Emily Temte, Amber Schemmel, Cristalyne Bell.

**Writing – original draft:** Jonathan L. Temte.

**Writing – review & editing:** Shari Barlow, Maureen Goss, Emily Temte, Amber Schemmel, Cristalyne Bell, Erik Reisdorf, Peter Shult, Mary Wedig, Thomas Haupt, James H. Conway, Ronald Gangnon, Amra Uzicanin.

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
