## [Decision Letter · Decision Letter 0]

11 Oct 2021

PONE-D-21-11837Cause-specific student absenteeism monitoring in K-12 schools for detection of increased influenza activity in the surrounding community - Dane County, Wisconsin, 2014-2020PLOS ONE

Dear Dr. Bell,

Thank you for submitting your manuscript to PLOS ONE. After careful consideration, we feel that it has merit but does not fully meet PLOS ONE’s publication criteria as it currently stands. Therefore, we invite you to submit a revised version of the manuscript that addresses the points raised during the review process. Please address well for the comments from the reviewers, especially for that related to study design.

We look forward to receiving your revised manuscript.

Kind regards,

Ka Chun Chong

Academic Editor

PLOS ONE

Journal Requirements:

2. Thank you for including your ethics statement:  "All components of this proposed study were reviewed and approved in writing by the Human Subjects Committees of the Education and Social/Behavioral Sciences IRB (initial approval on September 4, 2013; ID number: 2013-1268) and the University of Wisconsin Health Sciences-IRB (initial approval on December 5, 2013, with additional approvals as the protocol expanded and modified; ID number: 2013-1357). The study is in full compliance with the Health Insurance Portability and Accountability Act of 1996 (HIPAA), FERPA, and all other federally mandated human subjects regulations. The US Office of Management and Budget has approved all forms used in this study.".   

Please provide additional details regarding participant consent. In the ethics statement in the Methods and online submission information, please ensure that you have specified (1) whether consent was informed and (2) what type you obtained (for instance, written or verbal, and if verbal, how it was documented and witnessed). If your study included minors, state whether you obtained consent from parents or guardians. If the need for consent was waived by the ethics committee, please include this information.

3. In your Methods section, please provide additional information about the participant recruitment method and the demographic details of your participants. Please ensure you have provided sufficient details to replicate the analyses such as: 

a) a description of any inclusion/exclusion criteria that were applied to participant recruitment, 

b) a table of relevant demographic details of students

4. We note that Figure 2 in your submission contain map images which may be copyrighted. All PLOS content is published under the Creative Commons Attribution License (CC BY 4.0), which means that the manuscript, images, and Supporting Information files will be freely available online, and any third party is permitted to access, download, copy, distribute, and use these materials in any way, even commercially, with proper attribution. For these reasons, we cannot publish previously copyrighted maps or satellite images created using proprietary data, such as Google software (Google Maps, Street View, and Earth). For more information, see our copyright guidelines: http://journals.plos.org/plosone/s/licenses-and-copyright.

5. Thank you for stating the following in the Competing Interests section: "Dr. Jonathan Temte received in-kind research support from Quidel Corporation for the ORCHARDS study. Quidel Corporation did not direct or exert any influence over this manuscript."

We note that you received funding from a commercial source: "Quidel Corporation"

6. We note that you have indicated that data from this study are available upon request. PLOS only allows data to be available upon request if there are legal or ethical restrictions on sharing data publicly. For more information on unacceptable data access restrictions, please see http://journals.plos.org/plosone/s/data-availability#loc-unacceptable-data-access-restrictions. 

Reviewers' comments:

Reviewer's Responses to Questions

**Comments to the Author**

1. Is the manuscript technically sound, and do the data support the conclusions?

Reviewer #1: No

Reviewer #2: Yes

2. Has the statistical analysis been performed appropriately and rigorously? 

Reviewer #1: No

Reviewer #2: I Don't Know

3. Have the authors made all data underlying the findings in their manuscript fully available?

Reviewer #1: No

Reviewer #2: Yes

4. Is the manuscript presented in an intelligible fashion and written in standard English?

Reviewer #1: Yes

Reviewer #2: Yes

5. Review Comments to the Author

Reviewer #1: It is a pleasure to read this excellent manuscript, and I will be happy if my comments are helpful to improve the manuscript. Please find my specific comments below.

1. My main concern is regarding the methods of the paper. It says this is a prospective study. I am not sure how is it a prospective study. Would you please clarify it in the methods section?

2. Assuming it is a prospective study, is the same individual were repeatedly participating in the survey?

3. Assuming it is a prospective study, please provide descriptive statistics of the baseline, final wave and pooled in all waves.

4. Would you please report the regression results in tabular form?

5. Would you please run Zero-inflated negative binomial regression and compare with the existing regression results?

6. Would you please provide a figure for the flow of the sample and missing data analyses?

I hope my comments will be helpful.

Reviewer #2: This work deals with the assessment of the relation between students' absenteeism and influenza activity (epidemics) in the surrounding communities. The data rely on a 6 -year observation of student attendance in the Oregon school district employing Infinite Campus system. The main conclusion is that there is a correlation between absenteeism due to illness (a-I) and particularly influenza-like illness (a-ILI) and influenza spikes. This observation can be used to mitigate influenza outbreaks. The work is sound and well written and potentially publishable. However, I find the a-ILI lead time rather short to be useful, while the a-I correlation seems relatively weak. Of course it is an improvement over a-TOT, but I somehow miss a strong practical message of immediate applicability, which makes me wonder if a more specialized epidemics journal would not be a better place for this work. Perhaps the authors can strengthen this point,

-  In the abstract the authors suggest that monitoring a-ILI and a-I can provide early warning, but isn't a 1 day lead time too short for early warning? I suggest making this point clear.

- For the general audience of PLOS One, it will be useful if the authors briefly mention what kind of mitigation measures could be undertaken after a-I and a-ILI warning signals.

- What is the reason for decreasing a-ILI and increasing a-TOT with grades? Are they related to medical/social issues?

-  Perhaps I missed it, but were (some) students also ARI patients?

- Figs 4 and 5 shows that the a-ILI and MAI spikes nearly coincide, which agrees with the 1-day lead time, questioning whether this can be used as an early warning signal to mitigate influenza. I suggest to discuss this issue when discussing these figures and also in the Conclusions.

- Why did the authors expect that  dividing the data into K4 and K5-12 would have improved correlations?

6. PLOS authors have the option to publish the peer review history of their article (what does this mean?). If published, this will include your full peer review and any attached files.

Reviewer #1: No

Reviewer #2: No

---

## [Author Response · Author response to Decision Letter 0]

29 Dec 2021

Reviewer #1: 

1. My main concern is regarding the methods of the paper. It says this is a prospective study. I am not sure how is it a prospective study. Would you please clarify it in the methods section?

The reviewer appears to be envisioning a prospective cohort study, in which a single group of subjects is being followed over time, not the prospective monitoring of routinely collected school absence data. The full ORCHARDS methodology is long and complex and has been published as a stand-alone manuscript. We have emphasized the distinction between the data collection by the schools (the absenteeism data) and the home study data collection by research staff by adding the following at the beginning of the Materials and Methods section:

“ORCHARDS is a prospective, community-based study that captures school absenteeism data on a daily basis. In addition, school aged children with ARI are recruited to participant in a home study which allows evaluation of demographic, clinical, and virological correlates of respiratory infections.”

2. Assuming it [ORCHARDS] is a prospective study, is the same individual were repeatedly participating in the survey?

I am not sure what is meant here by the survey. We receive daily counts of absentee students in each of three categories of absenteeism. The data are anonymous. One would assume that some absent students are absent for more than one day, and, therefore, will have their absenteeism repeatedly recorded over more than one day. This constitutes the primary data set that is compared to daily counts of medically-attended influenza visits in surrounding communities using an independent surveillance system. There is no survey involved here.

Likewise, for our home study, we do allow for an individual to be visited more than once, but require this to be for a separate illness episode. Demographic, epidemiological, clinical and respiratory specimens are collected at the home visit. These data allow the validation of our definition of absenteeism due to influenza-like illness (a-ILI).

3. Assuming it is a prospective study, please provide descriptive statistics of the baseline, final wave and pooled in all waves.

The Wisconsin Department of Public Instruction determines each schools enrolled population once per year. As we point out in the text and Table 2, the student population in the Oregon School District grew marginally over the 6 years of the study (by about 1.3% per year). There are no missing data in the absenteeism data.

4. Would you please report the regression results in tabular form?

There are a large number of regression coefficients, none of which are directly interpretable. Unfortunately, given the brevity of the reviewer’s comment, it’s not really clear why they want a tabular presentation. Our statistician’s view was to possibly present the regression estimates at a regular series of timepoints, but that would likely be a very long table and would not be any more informative than the existing graphical representation of the model.

Accordingly, our statistician has generated a table presenting estimates from the GAM models (table included in response letter to reviewers; this could be included in the paper or provided as a supplement if so desired by the editor). For the calendar day effect, he has presented the estimated mean/CI for the number of events on the first Monday of each month (no estimates are presented for absences when school is closed, mostly July and August, but occasionally January and April). 

The MAI model/graphic actually nicely demonstrates how the model appropriately captures the very low rate of MAI outside of flu season, but the sporadic events outside flu season in most, if not all, years indicates that there’s never really a time when there’s zero risk of MAI.

5. Would you please run Zero-inflated negative binomial regression and compare with the existing regression results?

Our statistician has replied: “I strongly dislike zero-inflated models. They’re only appropriate in situations where it is impossible for the event of interest to occur on some days, but we don’t know exactly which days those are. The obvious application for our data would be if we didn’t know which days the schools were closed; there would be zero absences on those days for structural reasons, which could potentially produce a lot of excess zeros. Since we know when the schools were closed, we’ve already accounted for this. The other possibility would be that there are specific times when it is impossible, not just very unlikely, to have an event. I don’t think that applies to any of our outcomes. It’s possible to be absent, be absent due to illness, and be absent due to ILI at any time; you may be more likely to get ILI during flu season, but it’s not impossible to get an ILI (or even influenza) at any time.”

6. Would you please provide a figure for the flow of the sample and missing data analyses?

I hope my comments will be helpful.

We think this may reflect some confusion regarding the study design. There’s not a sample in the sense that the reviewer is talking about, and the only missing data issues would relate to participation in the home visits as there is no coupling between absenteeism and home visits. There are no missing data in the absenteeism data.

Reviewer #2: 

1. This work deals with the assessment of the relation between students' absenteeism and influenza activity (epidemics) in the surrounding communities. The data rely on a 6 -year observation of student attendance in the Oregon school district employing Infinite Campus system. The main conclusion is that there is a correlation between absenteeism due to illness (a-I) and particularly influenza-like illness (a-ILI) and influenza spikes. This observation can be used to mitigate influenza outbreaks. The work is sound and well written and potentially publishable. However, I find the a-ILI lead time rather short to be useful, while the a-I correlation seems relatively weak. Of course it is an improvement over a-TOT, but I somehow miss a strong practical message of immediate applicability, which makes me wonder if a more specialized epidemics journal would not be a better place for this work. Perhaps the authors can strengthen this point.

The “lead times” refer to above are the points at which correlations achieve maximal levels. We do comment that significant correlations extend to a lead time of at least 14 days, which is of great significance when considering the serial interval of influenza (see lines 298—300 and 311—313):

(298-300) “For all six seasons, however, significant, positive cross-correlations were achieved for a-I and a-ILI at least 14 days in advance of MAI (Figure 5).”

(311-313) “Moreover, as time frames are shifted, significant correlations are demonstrated between cause-specific absenteeism (a-ILI and a-I) and MAI well in advance of MAI in all cases.”

We would also argue that the correlations are reasonably high, and in the case of significant influenza outbreaks, very high. This underscores the potential value for identification of significant influenza outbreaks, such as highly virulent and/or pandemic strains (see lines 332—336):

(332-336) “The strength of correlations for individual outbreaks of influenza appears to follow the intensity of the influenza season. The most significant influenza seasons experienced by the OSD and the surveillance clinics (2017-2018 and 2019-2020) resulted in very high maximal correlations (r=0.64 and 0.82, respectively) between a-ILI and MAI. This supports potential benefit of cause-based absenteeism in monitoring systems for pandemic influenza.”

2. In the abstract the authors suggest that monitoring a-ILI and a-I can provide early warning, but isn't a 1 day lead time too short for early warning? I suggest making this point clear.

A one-day lead time only refers to the maximal correlation achieved (see lines 298—300 and 311—313), as above. Again, there is an interplay between the strength of correlation and the lead time. This paper provides the empirical data that can be used to formulate appropriate thresholds for triggering alert systems.

(298-300) “For all six seasons, however, significant, positive cross-correlations were achieved for a-I and a-ILI at least 14 days in advance of MAI (Figure 5).”

(311-313) “Moreover, as time frames are shifted, significant correlations are demonstrated between cause-specific absenteeism (a-ILI and a-I) and MAI well in advance of MAI in all cases.”

3. For the general audience of PLOS One, it will be useful if the authors briefly mention what kind of mitigation measures could be undertaken after a-I and a-ILI warning signals.

We have added content to the last paragraph in the conclusions following “thus allowing for mitigation efforts” to now read: “thus allowing for mitigation efforts, such as use of face masks, physical distancing, testing, isolation, quarantining, and enhanced vaccination programs.”

4. What is the reason for decreasing a-ILI and increasing a-TOT with grades? Are they related to medical/social issues?

The most likely reason is that younger children are more prone to develop fevers with ARIs and that older children are more likely to be absent for a wide variety of reasons. We did not attempt to assess the causation of this pattern.

5. Perhaps I missed it, but were (some) students also ARI patients? 

The absenteeism categories are for ILI (a subset of ARI), any illness, or any reason for absenteeism. ARI is requisite for a home visit, but ILI and absenteeism are not. The home visit component was used only to validate the definition of a-ILI and to assess the odds ratio of meeting a-ILI criteria if influenza is detected in a respiratory specimen. 

6. Figs 4 and 5 shows that the a-ILI and MAI spikes nearly coincide, which agrees with the 1-day lead time, questioning whether this can be used as an early warning signal to mitigate influenza. I suggest to discuss this issue when discussing these figures and also in the Conclusions.

See previous comments pertaining to the lead times and correlations above. 

7. Why did the authors expect that dividing the data into K4 and K5-12 would have improved correlations?

Younger children often have a more pronounced response to influenza and other respiratory viruses, than do older children, especially in regards to fever. This may be due to a lack of underlying, naturally acquired immunity. In addition, children from these five grades (out of 13) are represented in about 50% of our home visits. Finally, there is a natural cutoff in the Oregon School District as K-4 students are segregated into three schools, whereas the older age cohorts attend combined schools. We tested whether this age cutoff made any difference, so as to provide any additional guidance for districts that may implement this approach. It did not.

---

## [Decision Letter · Decision Letter 1]

16 Mar 2022

PONE-D-21-11837R1Cause-specific student absenteeism monitoring in K-12 schools for detection of increased influenza activity in the surrounding community - Dane County, Wisconsin, 2014-2020PLOS ONE

Dear Dr. Bell,

Thank you for submitting your manuscript to PLOS ONE. After careful consideration, we feel that it has merit but does not fully meet PLOS ONE’s publication criteria as it currently stands. Therefore, we invite you to submit a revised version of the manuscript that addresses the points raised during the review process.

We look forward to receiving your revised manuscript.

Kind regards,

Ka Chun Chong

Academic Editor

PLOS ONE

Journal Requirements:

Additional Editor Comments:

Please address well for the remaining comments.

Reviewers' comments:

Reviewer's Responses to Questions

**Comments to the Author**

1. If the authors have adequately addressed your comments raised in a previous round of review and you feel that this manuscript is now acceptable for publication, you may indicate that here to bypass the “Comments to the Author” section, enter your conflict of interest statement in the “Confidential to Editor” section, and submit your "Accept" recommendation.

Reviewer #2: All comments have been addressed

Reviewer #3: (No Response)

2. Is the manuscript technically sound, and do the data support the conclusions?

Reviewer #2: Yes

Reviewer #3: Yes

3. Has the statistical analysis been performed appropriately and rigorously? 

Reviewer #2: I Don't Know

Reviewer #3: Yes

4. Have the authors made all data underlying the findings in their manuscript fully available?

Reviewer #2: Yes

Reviewer #3: Yes

5. Is the manuscript presented in an intelligible fashion and written in standard English?

Reviewer #2: Yes

Reviewer #3: Yes

6. Review Comments to the Author

Reviewer #2: The authors have adequately responded to my concerns and, therefore, I can recommend the publication of this work.

Reviewer #3: Based on a high-quality database, this study examined the feasibility of using school absenteeism as an early alert signal to influenza in local communities. The data collection, preprocessing, and analysis are rigorous and clearly noted. The results showed the detailed cross-correlations between a-TOT, a-ILI, a-I absenteeism, and influenza cases in communities in consecutive influenza seasons from 2014 - 2020. The authors also discussed limitations and feasible mitigation measures according to the study results.

My suggestions for this study focus on the school absenteeism data.

1. It's reasonable that school-aged students are more vulnerable to influenza. Still, there is a limitation that the schools' Summer and Winter vacations last for months, which may lead to the missing data in these two specific periods and undermine the accessibility of alert.

2. Maybe it's better to list out the thresholds of a-I, a-ILI absenteeism numbers that represent the start of the influenza endemic in each year? It's just complementary. In real-world applications, it can help more compared to the correlation values.

7. PLOS authors have the option to publish the peer review history of their article (what does this mean?). If published, this will include your full peer review and any attached files.

Reviewer #2: No

Reviewer #3: No

---

## [Author Response · Author response to Decision Letter 1]

31 Mar 2022

Editor and Reviewer Comments: 

We have reviewed our reference list to ensure that it is complete and correct. In the previously submitted draft, we had removed one reference. In addition, we have updated the citation for [15] from the ePub date to the official date. There are no other changes to the current draft.

We appreciate the favorable assessment of reviewer #2 pertaining to our responses to concerns and recommendation for publication. We surmise that nothing else has been requested by Reviewer #2.

Likewise, we appreciate the very favorable assessment of reviewer #3 noting our high-quality database, the rigor and clarity of or data collection, preprocessing, analyses, and interpretation.

Regarding “a limitation that the schools' Summer and Winter vacations last for months, which may lead to the missing data in these two specific periods and undermine the accessibility of alert,” we agree that this does create a limitation, particularly during the prolonged summer months. Fortunately, influenza outbreaks are exceedingly uncommon during June, July and August in the Northern Hemisphere. We note that winter and spring breaks last no more than 16 and 9 days, respectively and have added the following to the paragraph on limitations:

“Fifth, absenteeism monitoring is only an option during times when schools are in session. As a consequence, gaps may occur during planned (e.g., winter and spring breaks) and unplanned breaks (e.g., weather closure) in the academic calendar. Winter and spring breaks are no longer than 16 and 9 days, respectively; unplanned closures tend to be of short (1-3 day) duration. Unusually timed influenza outbreaks during the long (3-month) summer break would be missed by absenteeism monitoring.” 

Regarding the suggestion to “list out the thresholds of a-I, a-ILI absenteeism numbers that represent the start of the influenza endemic in each year,” we have opted against this as it would likely provide a non-generalizable trigger level. The thrust of this study was to assess the detection of increased influenza activity in the surrounding community, not necessarily the onset. We are exploring the possible use of threshold values in a separate analysis, which is beyond the scope of the present assessment.

---

## [Editor Report · Decision Letter 2]

4 Apr 2022

Cause-specific student absenteeism monitoring in K-12 schools for detection of increased influenza activity in the surrounding community - Dane County, Wisconsin, 2014-2020

PONE-D-21-11837R2

Dear Dr. Bell,

We’re pleased to inform you that your manuscript has been judged scientifically suitable for publication and will be formally accepted for publication once it meets all outstanding technical requirements.

Kind regards,

Ka Chun Chong

Academic Editor

PLOS ONE
---

## [Editor Report · Acceptance letter]

8 Apr 2022

PONE-D-21-11837R2 

Cause-specific student absenteeism monitoring in K-12 schools for detection of increased influenza activity in the surrounding community - Dane County, Wisconsin, 2014-2020 

Dear Dr. Bell:

I'm pleased to inform you that your manuscript has been deemed suitable for publication in PLOS ONE. Congratulations! Your manuscript is now with our production department. 

Kind regards, 

on behalf of

Dr. Ka Chun Chong 

Academic Editor

PLOS ONE